# Fast evaluation of the adsorption energy of organic molecules on metals via graph neural networks

Sergio Pablo-García [1,2,3,10], Santiago Morandi [1,4,10], Rodrigo A. Vargas-Hernández[2,5], Kjell Jorner[2,3,6], Žarko Ivković [1], Núria López [1] ✉ & Alán Aspuru-Guzik [2,3,5,7,8,9] ✉

Modeling in heterogeneous catalysis requires the extensive evaluation of the energy of molecules adsorbed on surfaces. This is done via density functional theory but for large organic molecules it requires enormous computational time, compromising the viability of the approach. Here we present GAME-Net, a graph neural network to quickly evaluate the adsorption energy. GAME-Net is trained on a well-balanced chemically diverse dataset with $C_{1-4}$ molecules with functional groups including N, O, S and $C_{6-10}$ aromatic rings. The model yields a mean absolute error of 0.18 eV on the test set and is 6 orders of magnitude faster than density functional theory. Applied to biomass and plastics (up to 30 heteroatoms), adsorption energies are predicted with a mean absolute error of 0.016 eV per atom. The framework represents a tool for the fast screening of catalytic materials, particularly for systems that cannot be simulated by traditional methods.

Metal/organic interfaces are key to several fields including electronics, protective coatings and, in particular, heterogeneous catalysis[1]. Adsorption of organic species on metallic surfaces can be evaluated via density functional theory (DFT); this approach has been successfully applied to molecules containing up to one to six carbon atoms ($C_{1-6}$). However, DFT simulations become computationally expensive when dealing with: (1) large molecules with non-rigid bonds; (2) amorphous, partially disordered and/or polymeric structures; and (3) molecules with several conformations resulting in different bond patterns. Therefore, faster tools are needed to estimate the interaction of molecules derived, for instance, from plastics and biomass, but keeping the accuracy of DFT[2,3].

Large organic molecules can be seen as composed of different functional groups, and their structural information can be used to infer the molecular thermodynamic properties[4,5] through Benson's equation[6,7]:

$$T_{\mathrm{m}} = \sum_{i=1}^{N} T_i + c_{\mathrm{m}}, \tag{1}$$

where the thermodynamic property $T_{\mathrm{m}}$ of a molecule containing $N$ functional groups is obtained as the sum of each group contribution $T_i$ plus a constant $c_{\mathrm{m}}$ associated with the molecular property. Despite its simplicity, Benson's equation has an impressive accuracy for the formation energy of small gas-phase molecules such as hydrocarbons, alcohols and ethers, with errors lower than 0.05 eV (ref. 8). However, the description of radicals and strained rings requires additional corrections[9].

[1]Institute of Chemical Research of Catalonia, The Barcelona Institute of Science and Technology, Tarragona, Spain. [2]Department of Chemistry, University of Toronto, Lash Miller Chemical Laboratories, Toronto, Ontario, Canada. [3]Department of Computer Science, University of Toronto, Sandford Fleming Building, Toronto, Ontario, Canada. [4]Department of Physical and Inorganic Chemistry, Universitat Rovira i Virgili, Tarragona, Spain. [5]Vector Institute for Artificial Intelligence, Toronto, Ontario, Canada. [6]Department of Chemistry and Chemical Engineering, Chalmers University of Technology, Gothenburg, Sweden. [7]Department of Materials Science and Engineering, University of Toronto, Toronto, Ontario, Canada. [8]Lebovic Fellow, Canadian Institute for Advanced Research (CIFAR), Toronto, Ontario, Canada. [9]Acceleration Consortium, University of Toronto, Toronto, Ontario, Canada. [10]These authors contributed equally: Sergio Pablo-García, Santiago Morandi. ✉e-mail: nlopez@iciq.es; alan@aspuru.com

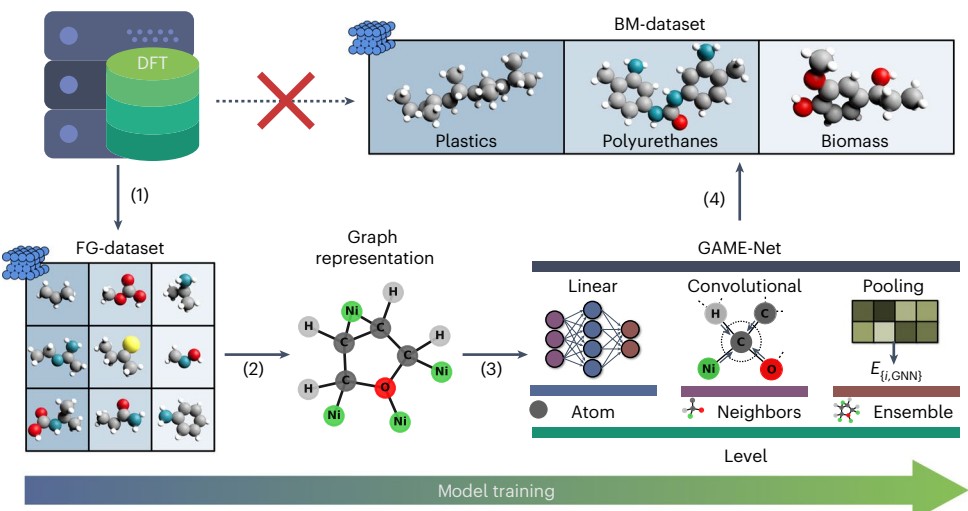

**Fig. 1 | Schematic illustration of the workflow for GAME-Net.** Starting from the DFT FG-dataset containing small adsorbates (3,315 points; step 1), we transform the sample adsorption systems to their corresponding graph representation (molecule and metal atoms directly interacting with the adsorbate) (step 2) to train the proposed GNN architecture (step 3). The final purpose is using GAME-Net to estimate the adsorption energy of big molecules $C_{<23}$ on metal surfaces present in the BM-dataset (step 4) avoiding the use of computationally expensive DFT calculations.

Attempts to transfer the Benson model to adsorption on metals have failed. This occurs because when molecules interact with surfaces, some internal bonds weaken and the corresponding density is responsible for the new bonds with the surface. This is known as the bond-order-conservation theory and was employed in early adsorption schemes[10]. Other additivity models derived from explicit DFT geometries and bond-counting techniques have faced some limitations[11–15]. However, a recent successful example of a group additivity scheme was presented for the determination of the free energy of 200 adsorbates and 151 reaction barriers in ethanol aqueous phase reforming on Pt, showing errors of around 0.12 eV (ref. 15). More recently, machine learning approaches[14,16–19], for instance, artificial neural networks (ANNs), have been introduced to obtain the adsorption energy of small $C_{1–3}$ fragments[20–22].

An alternative representation of molecules on surfaces is through graphs. The graph compresses the information of the atoms and the connectivity in a simple data structure, in an analog way to Benson's approach. Graph neural networks (GNNs), which are ANNs for the graph data structure type[23–26], have been successfully applied to predict chemical properties of molecules and materials[27,28]. For gas-phase molecules, GNNs have been able to predict molecular properties and their solvation energy with exceptional performance[29–31]. Extending to materials, specific convolutional graph layers can describe periodicity and predict the structure of metals and crystals[32,33]. For the adsorption of molecules on metals, GNNs have been used to estimate the DFT binding energy of small species (up to $C_2$ with O and N)[20] in the Open Catalyst Project[28,32,34–37]. Moreover, GNNs have been used to assess lateral adsorbate–adsorbate interactions[38]. Taking eight $C_{1–2}$ fragments on metals with different adsorbate–surface connectivity (1,422 points), a previous study[39] coupled a graph kernel to a Gaussian process regressor and obtained a reasonable performance for small molecules in a variety of metal and alloy surfaces (root mean square error 0.30 eV)[40]. Another study devised an algorithm to explore $10^8$ potential configurations of species with up to 6 carbon and oxygen atoms on 11 metals[19]. The workflow was built based on a graph enumeration platform, force field, DFT and machine learning, and was able to rapidly screen the stability of the configurations by introducing a fingerprint-like descriptor-based logistic regression[19]. As new GNN architectures trained to target additional chemical properties appear[41,42], finer-granularity graph models are emerging[43], creating a complete toolbox to address complexity.

In this article, we present GAME-Net (Graph-based Adsorption on Metal Energy-neural Network), a GNN model trained on an extensive DFT dataset consisting of closed-shell organic molecules (3,315 entries and common functional groups) adsorbed on transition metal surfaces, able to estimate the adsorption energy with an error comparable to DFT, using a simple molecular representation. GAME-Net can be used for predicting the adsorption energy of larger molecules derived from biomass, polyurethanes and plastics, allowing the study of chemical systems that are not amenable to DFT.

## Results

Our goal is to obtain the DFT ground-state energy of a closed-shell organic molecule on a metal surface, using the simplest graph representation. To this end, we followed the procedure illustrated in Fig. 1.

The workflow consists of the following steps: (1) generation and curation of the 'functional groups' (FG)-dataset, consisting of organic molecules with representative functional groups adsorbed on close-packed metal surfaces; (2) development of the graph representation for the adsorption and gas-phase systems from the DFT-optimized geometries; (3) design, training and testing a GNN model with the FG-dataset; and (4) assessment of the model performance with a dataset of larger molecules ('big molecules' (BM)-dataset, up to 22 carbon atoms) of industrial relevance including plastics, polyurethanes and biomass.

To start, we built the FG-dataset from scratch including 207 organic molecules adsorbed on 14 transition metals (Ag, Au, Cd, Co, Cu, Fe, Ir, Ni, Os, Pd, Pt, Rh, Ru and Zn) on their lowest-surface-energy facets. All generated computational data are available from the ioChem-BD repository[44,45]. The included molecules span the most common functional groups in organic chemistry containing N, O and S heteroatoms (Supplementary Section 1 and Supplementary Tables 1–9): (1) non-cyclic hydrocarbons; (2) O-functionalized (alcohols, ketones and aldehydes, ethers, carboxylic acids and carbonates); (3) N-functionalized (amines, imines and amidines); (4) S-functionalized (thiols, thioaldehydes and thioketones); (5) N- and O-functionalized combinations (amides, oximes and carbamate esters); and (6) aromatic molecules with up to two rings, also containing heteroatoms. Geometries were automatically generated and relaxed at the DFT PBE-D2 reparameterized for metals level following the rules described in Supplementary Section 2 and 'Automation of DFT data generation' in Methods. For each molecule, we sampled a number of rotational configurations and

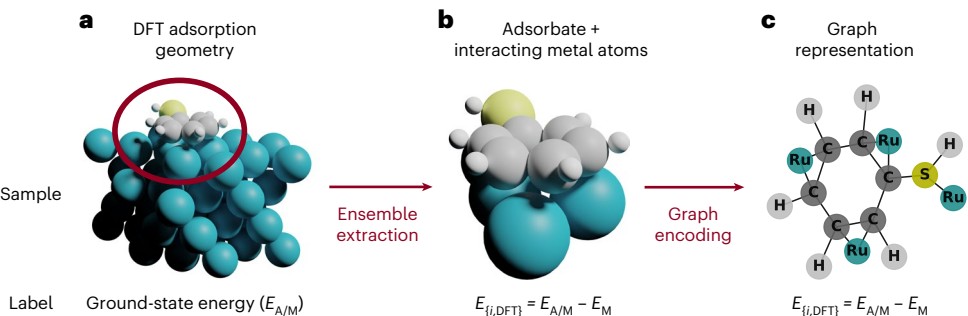

**Fig. 2 | Conversion from the adsorption geometry to graph. a**, Relaxed 3D structure with the total DFT energy $E_{A/M}$. **b**, Reduced 3D structure considering the bonds within the adsorbate (A–A) and between the metal atoms and the adsorbate (M–A), and the subtracted metal DFT energy, $E_{\{i,DFT\}} = E_{A/M} - E_M$. **c**, Graph codification of the reduced structure, keeping the label of the previous step.

different adsorption positions on the metal surface, retaining only the lowest-energy configuration.

## Graph representation

To build the graph to be used as input to the GNN, we started from the relaxed three-dimensional (3D) atomic coordinates. In the graphs, each atom is treated as a node storing its nature (element), while the bonds are taken as the edges. When interacting with the metal, the corresponding graph needs to incorporate the formed metal–adsorbate (M–A) bonds. We developed an algorithm to convert the adsorption structures into their corresponding graph representation. Figure 2 shows an example of the transformation process. Starting from the relaxed DFT geometry, we apply a modified version of the Voronoi tessellation algorithm[46].

The criterion to select the M–A bonds uses a slightly modified set of the atomic radii based on those of ref. [47]. Then, the graph representation contains all the molecule atoms and the metal atoms directly in contact with the adsorbate (M–A). We decided to keep our graph representation as simple as possible. The chemical element for each node is included in a one-hot encoding representation (Supplementary Section 2 and Supplementary Fig. 1), while the edges do not contain any chemical information of the bonds. The one-hot encoding is needed to convert categorical variables (as atomic elements) into machine-learning-suitable data structures. A set of filters is applied to the raw graph FG-dataset to avoid the presence of spurious representations during the training process (Supplementary Fig. 2). Detailed information regarding the conversion and curation procedure can be found in 'Graph representation algorithm' in Methods and Supplementary Fig. 3. To train the GNN, the generated graphs need to be labeled with their DFT energy. In this way, the energy of the total system $E_{A/M}$ (adsorbate on metal) is targeted, but this would imply accounting for the full metal graph, which is computationally heavy and complex to handle. Thus, we followed a Δ-ML approach[48]; the Δ learnt is the change in the energy of the molecule due the interaction with the metal. Our final target is the adsorption energy, $E_{ads}$, of the organic molecule, obtained as follows:

$$E_{ads} = E_{A/M} - E_M - E_A \qquad (2)$$

where $E_{A/M}$ is the energy of the adsorption system, $E_M$ the energy of the clean metal slab and $E_A$ the gas-phase molecule energy. Instead of this standard value, here we use a proxy energy, $E_{\{i,DFT\}}$, containing the first two terms:

$$E_{\{i,DFT\}} = E_{A/M} - E_M \qquad (3)$$

In this way, $E_{\{i,DFT\}}$ accounts for the energy of adsorbate $i$ and the perturbations caused by the bonding to the surface. This value is the target of GAME-Net ($E_{\{i,GNN\}}$). To get the adsorption energy, the energy of the gas-phase molecule should be subtracted.

## Model architecture

The adsorbed molecule graph can then be input into a neural network. GNNs are a type of ANN able to handle variable-size non-Euclidean data, represented as graphs. GAME-Net is built by assembling three building blocks: (1) fully connected layers, (2) convolutional layers and (3) a pooling layer. The first two blocks work at the node level. The fully connected layers apply a transformation to the embedding of the nodes. The convolutional layers exploit the graph connectivity, capturing the information from the embedding of the neighbors. The pooling layer allows the transformation from the node level to a graph-level representation predicting the energy, $E_{\{i,GNN\}}$. GAME-Net adopts GraphSAGE[49] as the convolutional layer and the Graph Multiset Transformer (GMT) as the pooling layer[50]. The architecture of GAME-Net has been defined by performing an extensive hyperparameter optimization study considering both architecture and training-related variables. Details about this process can be found in 'Hyperparameter optimization' in Methods. In total, GAME-Net has 285,761 parameters, and its detailed architecture is described in 'GAME-Net architecture' in Methods.

## Model performance

To estimate the generalization performance of GAME-Net with the FG-dataset in a robust way, we applied a fivefold nested cross-validation, performing a total of 20 learning processes with unique combinations of training, validation and test sets (more details in 'Model training' in Methods and Supplementary Figs. 5–9). For each model training, the cleaned graph FG-dataset is split into training, validation and test sets (60/20/20). The training set serves to learn the model parameters and the validation set is used to adjust the learning rate during the training process. The test set is unseen during the training process; thus it provides an unbiased evaluation of the GNN performance. As the FG-dataset is made up of subsets of specific chemical families (Fig. 3a), we implemented a stratified data splitting to ensure that all families are equally represented in each partition.

The nested cross-validation revealed a mean absolute error (MAE) of the predicted $E_{\{i,GNN\}}$ against the $E_{\{i,DFT\}}$ values of the test sets of 0.18 eV. Considering that the consensus error of DFT in adsorption is about 0.20 eV (ref. [51]) and that our values can be both above or below the 1:1 line, we conclude that the error of the method is similar to that of DFT itself. However, once trained with a sufficiently large and diverse dataset as the FG-dataset, the true advantage is the fast estimation of the DFT energy, which takes place on the order of milliseconds in a single central processing unit.

Figure 3d shows the error distribution grouped by chemical family. A similar standard deviation of 0.20 eV is found for each of the families (Supplementary Table 11). Lower MAEs are retrieved for amides

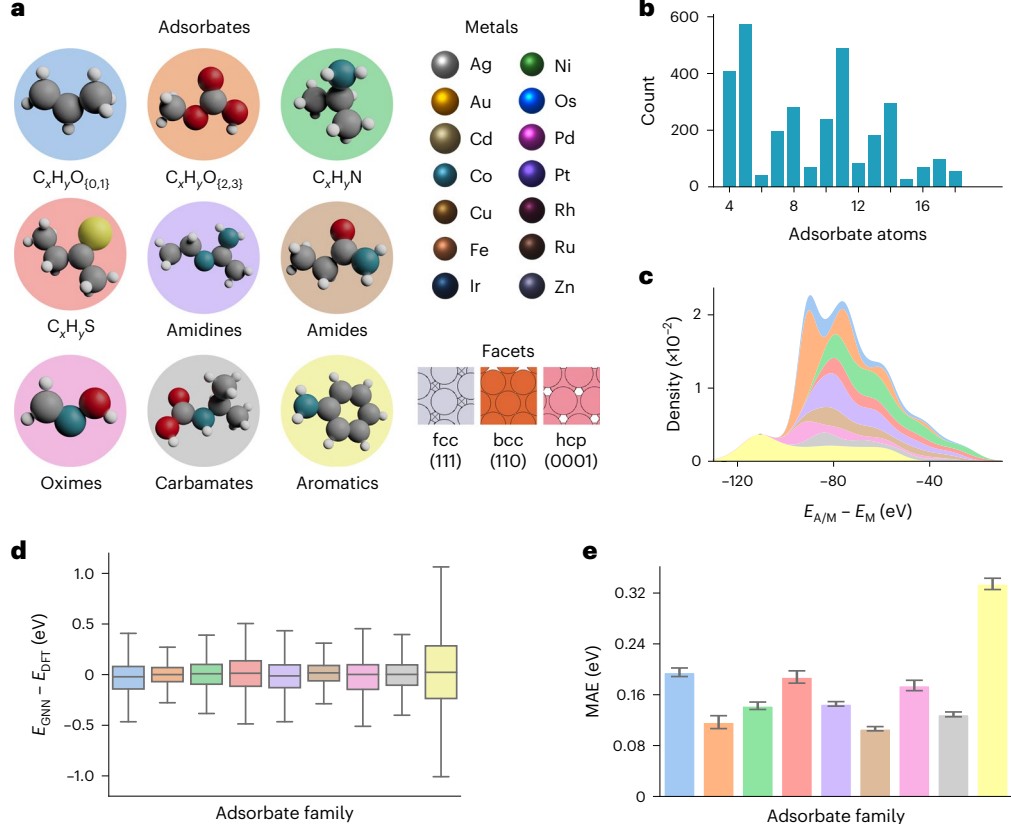

**Fig. 3 | GAME-Net training with the FG-dataset. a,** FG-dataset illustration, showing the included chemical families, metals and surface facets with the corresponding crystal system. fcc, face-centered cubic; bcc, body-centered cubic; hcp, hexagonal closed packed. **b,** Bar plot of the adsorbate atom count in the FG-dataset. **c,** Distribution of the DFT energy target $E_{A/M} - E_M$. **d,e,** Box plot of the error distribution (**d**) and the MAE (**e**) grouped by chemical family in the test sets from the fivefold nested cross-validation. The colors in **c**–**e** are associated with the chemical families in **a.** The number of data used for **d** and **e** is $n = 11,412$, due to the fact that the $k$-fold nested cross-validation involves including all the graph data of the FG-dataset ($n = 2,853$) in the test set $k - 1$ times, each with a different combination of training and validation sets for training the model. Each box plot in **d** defines the median as the box center, the interquartile range (IQR) as the box size, with whiskers extending for $1.5 \times$ IQR. Data in **e** are presented as mean ± s.e.m.

(0.11 eV), $C_xH_yO_{(2,3)}$ (0.1 eV), carbamates (0.13 eV) and oximes (0.18 eV). Larger errors are associated with the aromatic compounds, with an MAE of 0.34 eV. The source of this higher dispersion comes from conjugated rings, particularly those containing two rings. The error for aromatic compounds may be explained as due to the difficulties of the graph model to capture non-local electronic effects in aromatic rings and thus, additional work is needed to accurately represent these molecules, including ring identification routines in GAME-Net. Inherently non-local/non-additive effects have been also identified in localized coupled-cluster single–double methods[52,53]. Figure 3e shows the mean of the MAE among the different models generated during the cross-validation, grouped by family, and their associated standard error of the mean (s.e.m.) (Supplementary Table 11). Values obtained for the s.e.m. show that there is no substantial variation in the prediction performance among the models. Supplementary Fig. 10 and Supplementary Table 12 show the error distribution and the s.e.m. grouped by metal surfaces.

The inputs to the GNN are the optimized structures from the explicit DFT calculations in the FG-dataset. However, once GAME-Net is trained, an algorithm that docks a molecule on any of the investigated metal surfaces can be employed to generate an initial-geometry graph (not optimized) and an estimate of the energy can be extracted from GAME-Net. Although not trained on them, GAME-Net can assess different adsorption sites on the surface, that is, going from hollow sites, to bridge and top positions. We have randomly selected two molecules from each functional group and placed them on two different surface

sites (one from the original FG-dataset). We employed the initial geometry to get the corresponding graphs and compared the GNN energy prediction with the DFT energy after full relaxation. The results of this test (Supplementary Figs. 11–13) show that the graph representation is able to distinguish different adsorption sites, providing different graphs and consequently diverse model predictions. The mean absolute deviation of the GNN energy difference between two adsorption sites with respect to the DFT difference is 0.34 eV.

Similarly, we investigated the dependence of the adsorption energy as a function of the metal surface facet orientation, that is, structure sensitivity[54]. To this end, we have considered the (100) and (110) facets of the eight face-centered cubic (fcc) metals present in the FG-dataset and tested the generalization performance of the GNN with these unseen additional data (1,776 points for each facet), predicting their adsorption energy. The test revealed an MAE for the two facets of 0.34 eV and 0.41 eV, respectively, mainly affected by the poor performance for the aromatic molecules. The results can be found in Supplementary Figs. 14 and 15.

## Application to industrially relevant problems
The obtained GNN model generalizes the chemical patterns found within the FG-dataset, which presents functional groups that are commonly found as building blocks of more complex chemical structures. In this context, we tested GAME-Net on industrially relevant large molecules that are not amenable to DFT. The kind of challenges we decided to consider are the following: (1) molecules derived from biomass,

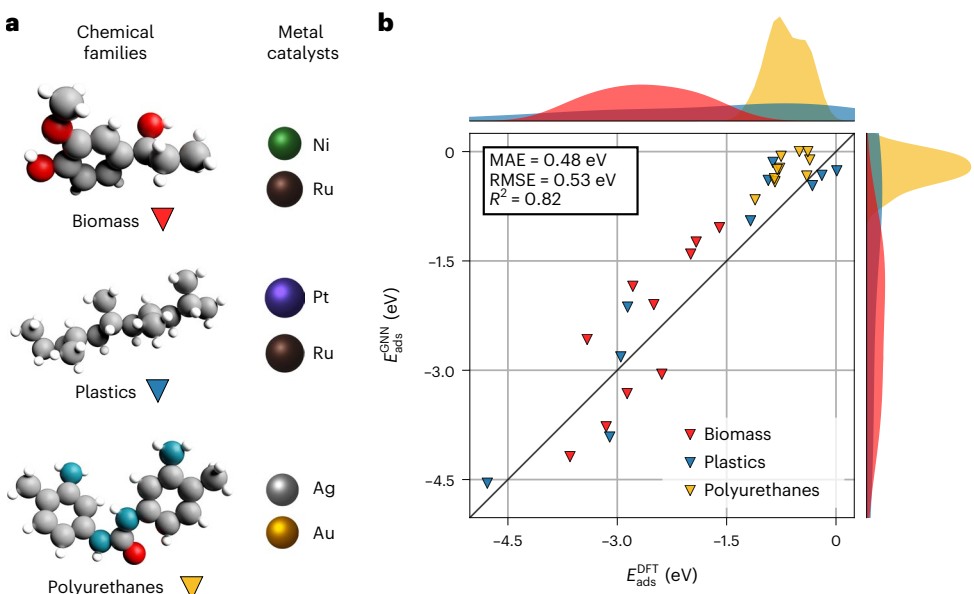

**Fig. 4 | GAME-Net application to the BM-dataset. a**, Chemical families present in the BM-dataset ($n$ = 30) used to evaluate the model performance outside the training data distribution. **b**, Parity plot for the adsorption energy of the samples in the BM-dataset predicted with GAME-Net, compared with the DFT values. RMSE, root mean square error.

(2) polyurethanes precursors derived from the 2,4-diaminotoluene and (3) polymeric molecules (for example, polyethylene, polyethylene terephtalate and polystyrene). Biomass-derived molecules typically contain multiple aromatic rings, unsaturated segments and O-containing functional groups. These molecules derive from the decomposition of lignin and the typical metals studied for its conversion are Ni and Ru[55]. Polyurethanes contain connected rings through N- and O-functionalized bridges, and their sustainable synthesis has been explored on Ag and Au catalysts[56]. Conversion of plastics is of great importance to society and developing the required technology for chemical recycling is an open challenge. However, DFT simulations have been of little help as these molecules are excessively large and consequently present a higher degree of flexibility. The metal catalysts considered for this family are Pt and Ru[57].

To test GAME-Net outside the training data distribution, we built the BM-dataset. The BM-dataset includes, for each mentioned group, five representative molecules in gas phase and adsorbed on two specific metals (Supplementary Section 3 and Supplementary Figs. 16–18). The optimized 3D structures were converted into graphs and used as test set for GAME-Net. Instead of considering $E_{\{i,DFT\}}$ as the final metric, in this case we consider $E_{ads}$, estimated by subtracting $E_{\{i,GNN\}}$ of the gas-phase molecule from $E_{\{i,GNN\}}$ of the metal-adsorption ensemble. Figure 4 shows the performance of the GNN model compared with the ground truth defined by DFT. The obtained global MAE for the BM-dataset is 0.48 eV, which is 0.016 eV per atom (Supplementary Table 13).

In general, the adsorption of large multifunctional molecules is reasonably estimated, even considering (1) the presence of aromatic rings in most of the adsorbates, and (2) the relatively small amount of data employed in the FG-dataset for training the model. Polyurethane precursors are well reproduced, even if their corresponding points are accumulated in the weak-binding-energy region (MAE = 0.43 eV). The adsorption of the plastics family is the best predicted, with MAE = 0.39 eV. The interaction of some of these molecules (polyethylene and polypropylene on Ru) with inert bonds is rather small and can be endothermic if the adsorbed conformation differs from the gas-phase one. Notably polyethylene terephtalate on Ru(0001), the biggest molecule (30 heteroatoms) in the BM-dataset with the highest adsorption energy (−4.78 eV), is one of the best reproduced of the entire dataset, with an error of 0.23 eV.

Finally biomass molecules, containing both rings and C−O functionalities, are also reproduced with a lower degree of accuracy (MAE = 0.63 eV), keeping, however, the relative ordering between the adsorption of the different molecules. The largest errors of the BM-dataset, which are associated with this family (0.94 eV and 0.83 eV), come from the same molecule, the only one in the dataset with an aromatic ring connected to a C $sp^2$ atom (Supplementary Fig. 16), a pattern for which GAME-Net has not been explicitly trained.

## Model benchmarking

An intrinsic feature of GAME-Net is that it targets the adsorption energy of closed-shell molecules. We tested the model with three external datasets including small open-shell fragments ranging from single atoms (C, H and O) to $C_4$ species[12,58,59] and we observed that the performance improves as the size of the considered fragments increases both within and between the datasets, as for these molecules the open-shell behavior becomes less dominant (Supplementary Section 4, Supplementary Table 14 and Supplementary Figs. 19–22).

To complement this, we benchmarked GAME-Net with two GNNs from the Open Catalyst Project, directional message passing neural network (DimeNet++) and polarizable atom interaction neural network (PaiNN)[28,36], using the FG-dataset as the training set. A detailed description of the benchmark study can be found in Supplementary Section 5, Supplementary Tables 15 and 16, and Supplementary Figs. 23 and 24. The original target of these models is the optimized adsorption energy starting from an initial structure and the full metal slab. In light of the difference between this setting and that defined for GAME-Net, we investigated the role of unrelaxed versus relaxed geometries and the number of metal atoms in the graph representation (full slab versus surface ensemble). The key learnings from the benchmark are: (1) the FG-dataset is robust, comprehensive and well balanced, enabling all benchmark models to provide satisfactory results; (2) the use of the graph representation based on the adsorption ensemble consistently yields superior results compared with the full slab for both DimeNet++ and PaiNN; and (3) GAME-Net performs similarly to DimeNet++ and PaiNN but at a substantially lower computational cost and time. In addition, while the benchmark models employ an initial 3D geometry to provide a final optimized energy, GAME-Net is more intuitive as the guess structures are distilled as graphs and thus the chemistry

becomes transparent in the interpretation of the bond at the interface. This is particularly evident when assessing the adsorption energy on different surface positions (top versus hollow sites), where we can trace completely the type of bond, essential information for experimental scientists to understand reactivity.

## Discussion

GAME-Net demonstrates its robustness when explaining the chemical adsorption. As functional groups in the dataset present a wide variety, GAME-Net generalizes to bigger compounds, providing a fast alternative to evaluate the adsorption energy. Each structure in the FG-dataset required on average 5.4 h to be evaluated. These are small calculations on relatively small unit cells; thus they can be run in modest computational facilities. The large molecules of industrial interest still would require massive computational resources due to the large unit cells required and sequential DFT evaluations in powerful machines. The total elapsed time needed to generate the FG-dataset was 16,617 h, and a single GNN training in a normal laptop graphics processing unit takes around 5 min. The hyperparameter optimization involved thousands of learning processes, requiring the use of a computational node equipped with high-performance-computing-grade graphics processing units and contributing importantly to the final cost. As the actual output of GAME-Net is not the adsorption energy but the scaled system energy (equation (3)), getting the adsorption energy prediction for a specific system requires two passes through the GNN: one for the adsorption graph and one for the graph of the gas-phase molecule (or alternatively its DFT value as this one is relatively cheap). For a comparative DFT versus GAME-Net evaluation, the time gain is several orders of magnitude (7–9 depending on the size of the molecule), while the gray hidden cost of training the network is $10^7$–$10^8$ s. The positive part is that once trained, GAME-Net can be widely applied to other compounds without the need for external computational resources, as it can run on a normal laptop in milliseconds.

Compared with other methods in the literature based on linear-scaling relationships, group additivity models or earlier graph representations, our model has been devised to provide results for a wider and more complex chemical space, where the functional groups are not required to be spatially isolated in the molecule (not-interacting) and more than one can interact with the surface[15,19,60] (Supplementary Fig. 25).

The simplicity of our method might present some limitations when considering complex catalysts with multiple components or phases. Our graph representation considers only the first metal neighbors between the adsorbate and the surface, disregarding potential different metal atoms not directly bonded to the adsorbate (alloys), changing coordination number or stepped surfaces[35,54]. In particular, the surface coordination dependence has been tested for only low-energy surfaces (fcc (110) and (100)), and for very low-coordinated surface sites with complex topography, important deviations could be observed. Another degree of freedom that the model has not been trained with is strain, and thus the response in this case cannot be anticipated. As for alloys and dopants, the model is probably able to capture the electronic contributions but less so the local strain effects. In addition, in complex catalysts, secondary phases and supports might contribute or change adsorption patterns. Finally, simplifications regarding the lateral effects of the adsorbed molecules, such as coverage effects and solvation, are also being investigated with neural networks; therefore, they could potentially be implemented in a similar manner[38].

The limitations can be seen as strengths, as they increase the generality of the model. A method that takes more information into account will become more specialized and difficult to deploy. The overall simplicity of our methodology relies on a few key aspects: (1) the robustness coming from the wide chemical span covered owing to the FG-dataset used to train GAME-Net, (2) the possibility to use guess initial structures to obtain the prediction, and (3) the availability

to run GAME-Net on a consumer laptop and obtain the prediction in milliseconds.

In summary, we have generated a robust, well-balanced and chemically diverse dataset, including all the relevant functional groups in chemistry consisting of $C_{1-10}$ molecules, for their adsorption on closed-packed metal surfaces. The dataset is used to train the proposed GNN architecture. The fivefold nested cross-validation revealed an MAE of 0.18 eV when applied to the FG-dataset. Once trained, the time required to obtain the energy estimation from GAME-Net is at least six orders of magnitude lower than that for DFT. Application of GAME-Net to larger molecules related to biomass conversion, polyurethane synthesis and plastic chemical recycling shows the potential of geometric deep learning models to reach areas that can not be easily addressed by standard first-principles techniques. Our work provides a tool for building graph-based frameworks capable of learning complex chemical patterns from high-quality datasets composed of small molecules.

## Methods

### Density functional theory

We performed the DFT simulations with the Vienna Ab-initio Simulation Package, VASP 5.4.4 (ref. 61). The FG-dataset used to train GAME-Net consists of 207 closed-shell organic molecules both in the gas phase and adsorbed on 14 metals (3,315 samples). The functional of choice was the Perdew–Burke–Ernzerhof (PBE)[62] with D2 (ref. 63) and our re-parameterized values for metals[64]. Core electrons were represented by projector-augmented-wave pseudopotentials and valence electrons were represented by plane waves with a kinetic energy cut-off of 450 eV. Electronic convergence was set to $10^{-5}$ eV and atomic positions were converged until residual forces fell below 0.03 eV Å$^{-1}$. The gas-phase molecules were relaxed in a cubic box with 20 Å sides. Metals include eight fcc, one body-centered cubic and five hexagonal closed packed, and (111), (110) and (0001) surfaces, respectively (Ag, Au, Cd, Co, Cu, Fe, Ir, Ni, Os, Pd, Pt, Rh, Ru and Zn). The calculated lattice parameters for the metals show good agreement with experimental values. Extension to other surface orientations was performed on the same molecules adsorbed on the (100) and (110) facets of fcc metals, not included in the FG-dataset. These additional test sets have 1,776 data each[65]. Metal surfaces were modeled by four-layer slabs, where the two uppermost layers were fully relaxed and the bottom ones were fixed to the bulk distances. Only the fcc(110) facets were modeled with six layers, keeping the 3 at the bottom fixed. The FG-dataset samples were generated on $2\sqrt{3} \times 2\sqrt{3} - R30°$ supercells (all except the Fe and Co samples), with a surface coverage concentration of 0.02 molecules per square ångström, a reasonable value to avoid lateral interactions. We employed a set of rules to obtain the optimum adsorption site for the molecules on the surface (Supplementary Section 2). These include the rotation of several bonds in the molecular skeleton and the sampling of different adsorption positions. The vacuum between the slabs was set larger than 13 Å and the dipole correction was applied in the $z$ direction[66]. The Brillouin zone was sampled by a Γ-centered $3 \times 3 \times 1$ $k$-points mesh generated through the Monkhorst–Pack method[67]. Regarding the BM-dataset, the samples belonging to the plastics family were calculated on the supercells of the FG-dataset multiplied twice in the $x$–$y$ directions to make the larger molecules fit into them, while the slabs for the polyurethanes and biomass were retrieved from ref. 56 and ref. 55, respectively.

### Automation of DFT data generation

Building a proper dataset for machine learning purposes requires the automation of the data-generation process[68]. To build and obtain the adsorbate–metal combinations included in the FG-dataset, we executed the following procedure. (1) First, the metallic surfaces were built starting from their respective bulks. (2) Then the SMILES[69] of the molecules in the FG-dataset were fed to Open Babel to generate the .xyz geometry files of the molecules, applying the MMFF94 force field[70]. (3) The .xyz geometry files were converted with Pymatgen 2022.11.7

(ref. [71]) to VASP POSCAR files representing the molecules in cubic cells with a dimension of 20 Å. The gas-phase molecules were calculated with this initial geometry. (4) The relaxed molecules were copied and placed at a distance of 2.2 Å from the Rh slab, our reference metal, following Supplementary Section 2. (5) The resulting structures were relaxed with VASP. Once full convergence was reached, the geometry of the relaxed molecules on Rh was extracted and automatically placed on the slabs of the all the other metals preserving the adsorbate distance from the Rh slab. (6) The obtained structures were checked for conformation errors (less than 1%), and if needed, manually built and relaxed again. Typical problems encountered were related to the fragmentation of the adsorbate during the relaxation due to the unstable initial geometry.

For the BM-dataset, the molecules were manually built, adsorbed on the metal surfaces and relaxed using VASP, including the gas-phase molecules. Structures obtained from both datasets were uploaded to ioChem-BD[44,45].

### Graph representation algorithm

To convert the 3D structures obtained from DFT to their respective graph representation, we applied a modified version of the algorithm presented in ref. [46], implemented in pyRDTP 0.2. To define the neighbors of each atom, the algorithm reads the relaxed 3D atomic positions from the geometry file (CONTCAR) generated by VASP, and applies the Voronoi tessellation method. This method creates a partition of the 3D space, defining a region for each atom that consists of all points of the space closer to that atom than to any other. Two atoms are considered connected if they share a Voronoi facet and their distance is less than the sum of the atomic covalent radii plus a tolerance distance. In this work, we used as covalent radii those provided by ref. [47] multiplied by a scaling factor of 1.5 for metals, and a tolerance of 0.5 Å to help in the detection of metal–adsorbate connections.

Once the connectivity is defined, the graphs are generated, representing the atoms as nodes and the detected connections as edges. The metal atoms not directly connected to the adsorbate are not considered during the graph-generation process. The atomic elements are embedded to the nodes using the one-hot encoding approach as implemented in scikit-learn[72] (Supplementary Fig. 1). This step is needed to convert categorical variables (as atomic elements) into machine-learning-suitable data structures. This algorithm is applied to all the samples in both the FG-dataset and the BM-dataset. For a fraction of the adsorption systems (about 4%), the graph conversion results in inaccurate representations, as for specific geometries the M−A distance is so high that the algorithm is unable to properly define their connectivity. This is due to the fact that the algorithm is based on a purely geometric criterion that defines the edges relying on a set of empirical covalent radii, which are fixed for each element and do not account for all the possible phenomena occurring in catalytic systems. Thus, the strategy for minimizing the amount of bad representations has been to fine tune the tolerance parameter and the covalent radii scaling factor. To discard inaccurate graph representations and properly curate the graph dataset to be suitable for the model training, we implemented the following four sieves (Supplementary Fig. 2): (1) a first filter that discards the graphs representing adsorption configurations without the presence of metal atoms; (2) a second filter that verifies the correct connectivity of C and H atoms within the molecules−connectivity of carbon atoms is properly defined if the number of edges connecting it to other atoms in the molecule is equal or less than 4, while hydrogen atoms are correctly connected if its number of edges is exactly one (our molecules are closed shell and thus the interaction with the metal is of van der Waals type at most); (3) a filter to prevent the inclusion of DFT samples containing more than one adsorbate on the slab or with a final geometry in which the adsorbate has dissociated; and (4) a last filter for removing duplicate graphs deriving from the presence of stereoisomers adsorbed in the same configuration on the metal surface.

The amount of graphs pruned out after the first two filters intrinsically depends on the graph conversion algorithm: a higher applied tolerance would reduce the number of discarded graphs after the first filter, but at the same time it increases the number of removed graphs after the second filter, due to the creation of non-physical connections within the adsorbates.

### Hyperparameter optimization

GAME-Net design involved a series of hyperparameter optimization studies to explore the vast space defined by all the variables affecting the final model performance. The hyperparameters are the variables that are not trainable parameters, but that affect model performance to the same extent. These can be divided in two groups, training-related hyperparameters (that is, learning rate, optimizer, batch size and so on) and model-related hyperparameters, which define the model architecture (that is, activation function, layer depth and width, bias and so on). We adopted the hyperband asynchronous algorithm (ASHA)[73] implemented in RayTune[74]. ASHA combines random search and aggressive early stopping to optimize the hyperparameters and is based on the proved fact that to find the best hyperparameter setting, just a small number of iterations (epochs) is sufficient to discriminate between poor and promising candidates. The hyperparameter space, shown in Supplementary Table 10, was investigated by randomly picking 10,000 different settings and running model training for each using ASHA, with a grace period of 15 epochs (for example, the worst models are discarded after training them for a minimum of 15 iterations) and a maximum of 200 for the best ones. The final adopted hyperparameter setting is the one that minimizes the MAE of the energy of the samples belonging to the BM-dataset.

### GAME-Net architecture

The architecture of GAME-Net developed in this work is shown in Supplementary Fig. 4. The input graphs are represented by a set of node feature vectors, each of them being a 19-dimensional array (14 metals + C, H, O, N and S) needed for representing the chemical element via the one-hot encoder, and by the graph connectivity in coordinate format (Supplementary Fig. 1). To transform the graph representation into the DFT energy prediction of the associated system, the following transformations are applied in the listed order.

(1) First, each node feature vector is transformed via one dense layer, which increases the dimensionality of the vector from 19 to 160, $\mathbb{B}^{19} \rightarrow \mathbb{R}^{160}$, where $\mathbb{B}$ denotes the Boolean space used to define the nodes via one-hot encoding. The bias term is not present in the linear transformation.

(2) Three GraphSAGE[49] convolutional layers are applied to all the nodes to capture the information from the neighbors by exploiting the graph connectivity, $\mathbb{R}^{160} \rightarrow \mathbb{R}^{160}$.

(3) Finally, the information embedded in the nodes is compressed into a graph representation via the GMT[50], a global pooling layer that returns back a scalar value, namely, the DFT energy prediction of the chemical system represented by the initial input graph. $\mathbb{R}^{N \times 160} \rightarrow \mathbb{R}$, where $N$ is the number of nodes in the graph.

The activation function used in the node-level layers (all except the pooling layer) is the rectified linear unit (ReLU)[75]. GAME-Net has been trained with 2,853 graphs, ending up with a GNN that contains 285,761 trainable parameters, 129,121 of them (45%) belonging to the GMT pooling layer due to its internal complexity, and the remaining parameters equally distributed among the other layers. The number of parameters employed in the GNN probably contains some redundancies; however, eliminating those might be a more complex task than employing the compact structure of GAME-Net.

## Model training

The training processes have been performed by minimizing the MAE as a loss function with the ADAM optimizer[76]. The learning rate value is steered with the reduce-on-plateau scheduler. The initial learning rate was set to $10^{-3}$ and is reduced exponentially by the scheduler every time in which there is no improvement after 5 epochs (patience) in the MAE of the validation set. The minimum learning rate possible was set to $10^{-8}$, while the decrease factor was set to 0.7. In each training, 200 epochs were performed. During each epoch, the training set is fed to the model in batches of 16 samples, performing a backward propagation and updating the model parameters after each batch.

GAME-Net performance was assessed by applying a stratified splitting of the FG-dataset by chemical families followed by a fivefold nested cross-validation. The first (Supplementary Fig. 5) allows a proper distribution of all the chemical families among the splits, while the latter provides a robust estimation of the GNN generalization performance. The nested cross-validation approach follows the process depicted in Supplementary Fig. 6: after the partition in five stratified splits, each split is employed as test set; for each split used as a test set, four splits are left and each is employed as a validation set. This leads to a final validation approach consisting of 20 independent learning processes, each performed with a unique combination of the splits among training, validation and test sets. The generalization performance of the model is finally assessed by averaging the MAE of the absolute errors of the learning processes performed using the same test set. Supplementary Fig. 7 shows the typical trend of MAE and learning rate during a model training, together with the graph target distribution in the training, validation and test sets after the stratified data splitting. It is important to mention that after the creation of the training, validation and test sets in each learning process, a standardized target scaling is applied using the mean and standard deviation of the energy values of the samples from the training and validation sets, discarding the samples from the test set as its inclusion would lead to data leakage. The target scaling is essential to ensure a stable learning process: if the target variable in the graph dataset has a large spread of values, it may result in large error gradients causing model parameters to change dramatically, making the learning process unstable.

## Computational tools

GAME-Net was built with PyTorch Geometric 1.6 (ref. 77) running over PyTorch 1.10 (ref. 78). PyRDTP 0.2 (ref. 79) (gitlab.com/iciq-tcc/nlopez-group/pyrdtp) and NetworkX 2.6 (ref. 80) were used to convert the molecular structures into graphs. Stratification and cross-validation algorithms were developed exclusively for this work and implemented in Python 3.9 (ref. 81). Matplotlib 3.5.1 (ref. 82) and Seaborn 0.11.2 (ref. 83) were used to draw the plots presented in the paper.

## Data availability

The DFT datasets are available in the ioChem-BD repository[44]. Source data for Figs. 3 and 4, and the Supplementary figures containing data plots are available with the paper. A simplified version of the used datasets, containing the DFT geometries and energies, is available with the code repository in Zenodo[84].

## Code availability

The Python code framework has been publicly released under a MIT license and is available at gitlab.com/iciq-tcc/nlopez-group/gnn_eads. The version of GAME-Net used in this work (0.2.0) has been uploaded to Zenodo[84].

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

## Acknowledgements

This work was supported by Ministerio de Ciencia e Innovación, with ref. no. PID2021-122516OB-I00 and with a Mobility Grant within Severo Ochoa MCIN/AEI/10.13039/501100011033 CEX2019 000925 S. This publication was created as part of NCCR Catalysis (grant number 180544), a National Centre of Competence in Research funded by the Swiss National Science Foundation. We thank BSC-RES for generously providing computational resources and Generalitat de Catalunya for support (2021 SGR 01155). K.J. acknowledges funding through an International Postdoc grant from the Swedish Research Council (no. 2020-00314). Ž.I. thanks Fundació la Caixa for a Summer Fellowship. A.A.-G. acknowledges support from the Canada 150 Research Chairs Program as well as Anders G. Fröseth. We thank M. Álvarez-Moreno for software support.

## Author contributions

S.P.-G.: software, methodology, investigation, benchmarking and writing—original draft. S.M.: software, methodology, investigation, data generation and curation, model design and writing—original draft. R.A.V.-H.: methodology, investigation, reviewing and editing. K.J.: methodology, reviewing and editing. Ž.I.: data generation. N.L. and A.A.-G.: conceptualization, funding acquisition, supervision, project administration and writing—review and editing.

## Competing interests

The authors declare no competing interests.

## Additional information

**Correspondence and requests for materials** should be addressed to Núria López or Alán Aspuru-Guzik.

