## [Peer Review File · Nature Computational Science]

Peer Review Information

Journal: Nature Computational Science

Manuscript Title: Fast Evaluation of the Adsorption Energy of Organic Molecules on Metals via Graph Neural Networks

Corresponding author name(s): Núria López, Alán Aspuru-Guzik

Reviewer Comments & Decisions:

Decision Letter, initial version:

Date: 18th October 22 10:22:07

Last Sent: 18th October 22 10:22:07

Triggered By: Kaitlin McCardle

From: kaitlin.mccardle@us.nature.com

To: nlopez@iciq.es

BCC: kaitlin.mccardle@us.nature.com

Subject: Decision on Nature Computational Science manuscript NATCOMPUTSCI-22-1074

Message: ** Please ensure you delete the link to your author homepage in this e-mail if you wish to forward it to your co-authors. **

Dear Professor López,

Thank you again for submitting your manuscript "Fast Evaluation of the Adsorption Energy of Organic Molecules on Metals via Graph Neural Networks". I am pleased to tell you that we are sending your paper out for formal peer review. Before we can do so, please read the below carefully as we require a few documents.

If you have not done so already, please alert us to any related manuscripts from your group that are under consideration or in press at other journals, or are being written up for submission to other journals (see <https://www.nature.com/authors/policies/duplicate.html> for details).

We are asking all corresponding authors of primary research articles to complete an Editorial Policy Checklist that verifies compliance with all required editorial policies. Please note that the form is a dynamic 'smart pdf' and must therefore be downloaded and completed in Adobe Reader. We will then flatten them for ease of use by the reviewers. If you would like to reference the guidance text as you complete the template, please access these flattened versions at <https://www.nature.com/authors/policies/availability.html>

Editorial Policy Checklist: <https://www.nature.com/documents/nr-editorial-policy-checklist.zip>

In addition, as your paper relies on code that is central to the main claims, we will ask the reviewers to evaluate the code during the peer review process (for more details on this please see this editorial in Nature <https://www.nature.com/articles/d41586-018-02741-4>). For your paper, this means you'll need to share all code/data related to experiments as well as methodology.

Reproducibility and re-usability of code are very important to us so, to facilitate this process, we are currently running a trial in partnership with Code Ocean to enable authors to share fully-functional and executable code accompanying their articles and to facilitate peer review of code by the reviewers (for more details please see <http://blogs.nature.com/ofschemasandmemes/2018/08/01/nature-research-journals-trial-new-tools-to-enhance-code-peer-review-and-publication>). We expect this functionality to speed up the peer review of your paper as it will facilitate the reviewer's assessment.

The use of the Code Ocean platform for peer review of code associated with this paper will be under the same confidentiality and anonymity agreements as the rest of manuscript materials.

Code Ocean is a cloud-based reproducibility platform where authors upload code and data and configure the necessary computational environment for reproduction. The code, data, metadata, and computational environment -- called a 'compute capsule' -- can then be accessed by reviewers in an anonymous fashion, and upon publication, provided to readers via a link from the article. Code Ocean supports all open source programming languages, as well as Stata and MATLAB and compute capsules can be created from existing GitHub folders by easy drag and drop.

Code Ocean staff will assist you in generating a compute capsule for your code and a working copy of this compute capsule will be used in the peer review process (after a brief review by Code Ocean staff, to ensure that everything runs). If you have selected Double Blind Peer Review, we will make sure the capsule contains no information about your identity.

If your code is accepted for publication, Code Ocean will assign a Digital Object Identifier (DOI) to your compute capsule. It will then be embedded into your article. Code Ocean, through CLOCKSS, will guarantee the preservation of all elements of Code Ocean's compute capsules, including the code, data, results, metadata, Dockerfile and Docker image (computational environment) associated with your paper.

By using this platform, other researchers will then be able to easily find and run the code, as well as build upon your work, without any additional setup or configuration of the software. It will also enable preservation of your code, data and the complete environment so that the code associated with this publication is maintained. Should the paper be rejected, you will retain full control over the compute capsule, and be able to decide what to do with it (publish it, modify it etc).

We very much hope you will be interested in engaging in this trial. Please let us know as soon as possible if you wish to participate and we will provide you with further guidelines for setting up the compute capsule. An overview of the process can be found here: <https://help.codeocean.com/publishing-on-code-ocean/peer-review-on-code-ocean> .

Alternatively, if you do not want to engage in this trial, or if Code Ocean is not a good fit (<https://help.codeocean.com/en/articles/3294415-what-is-and-is-not-a-good-fit-for-publishing-and-sharing-on-code-ocean>), we ask you to complete the following Software and custom code submission checklist:

Software supplement: <https://www.nature.com/documents/nr-software-policy.pdf>
(Please note that the form is a dynamic 'smart pdf' and must therefore be downloaded and completed in Adobe Reader.)

To improve transparency in authorship we are requesting that all authors identified as 'corresponding author' on published papers create and link their Open Researcher and Contributor Identifier (ORCID) with their account on the Manuscript Tracking System (MTS), prior to acceptance. ORCID helps the scientific community achieve unambiguous attribution of all scholarly contributions. You can create and link your ORCID from the home page of the MTS by clicking on 'Modify my Springer Nature account'. For more information please visit www.springernature.com/orcid.

Please note that *Nature Computational Science* is a Transformative Journal (TJ). Authors may publish their research with us through the traditional subscription access route or make their paper immediately open access through payment of an article-processing charge (APC). Further information regarding *Nature Computational Science* publishing options and our APC is available [here](https://www.springernature.com/gp/open-research/policies/journal-policies).

For submissions from January 2021, if your paper is accepted for publication in *Nature Computational Science*, you will be asked to choose the publishing option that works best for you. If your research is supported by a funder that requires immediate open access (e.g. according to Plan S principles) then you should select the gold OA route, and we will direct you to the compliant route where possible. For authors selecting the subscription publication route our standard licensing terms will need to be accepted including our self-archiving policies. Those standard licensing terms will supersede any other terms that the author or any third party may assert apply to any version of the manuscript.

Finally, we encourage you to share a preprint of the original submitted version of your paper so as to minimize delays in communicating your research findings; benefits of preprints include early visibility, and citations (<https://www.natureindex.com/news-blog/preprints-boost-article-citations-and-mentions>) and demonstration of research progress. You may want to consider the multidisciplinary Research Square preprint platform (<https://www.researchsquare.com/browse>), provided by our partner Research Square, where your preprint will be publicly available with a citable DOI under a CC-

BY license. You are of course free to use a discipline-specific preprint platform of your choice. More information about our preprint policy can be found in the following link: <https://www.nature.com/nature-research/editorial-policies/preprints-and-conference-proceedings#preprints>

Please use the following link to submit the required checklists; please also resubmit your original manuscript files, or revised versions of them as a result of filling out the checklists:

[REDACTED]

** This url links to your confidential homepage and associated information about manuscripts you may have submitted or be reviewing for us. If you wish to forward this e-mail to co-authors, please delete this link to your homepage first. **

Please note that Nature Computational Science implements transparent peer review of original research manuscripts, in which we publish the reviewer comments to the authors, author rebuttal letters and editorial decision letters as a supplementary peer review file, if the author agrees at the point of acceptance. This will apply to new manuscripts submitted on or after 17th Feb 2021. Upon author request, confidential information and data can be removed from the reviewer reports and rebuttal letters prior to publication. For more information, please refer to our [FAQ page](https://www.nature.com/documents/nr-transparent-peer-review.pdf).

Thank you very much for your attention to this. We look forward to hearing from you, about the Code Ocean trial and to receiving your editorial policy checklist, but please let me know if you have any questions.

Best regards,
Kaitie

--

Kaitlin McCardle, PhD
Associate Editor
Nature Computational Science

Author Rebuttal to Initial comments

Not Applicable

Decision Letter, first revision:

Date: 28th November 22 14:23:33

Last Sent: 28th November 22 14:23:33

Triggered By: Kaitlin McCardle

From: kaitlin.mccardle@us.nature.com
To: nlopez@iciq.es
BCC: kaitlin.mccardle@us.nature.com
Subject: Decision on Nature Computational Science manuscript NATCOMPUTSCI-22-1074A
Message: ** Please ensure you delete the link to your author homepage in this e-mail if you wish to forward it to your co-authors. **

Dear Professor López,

Your manuscript "Fast Evaluation of the Adsorption Energy of Organic Molecules on Metals via Graph Neural Networks" has now been seen by 3 referees, whose comments are appended below. You will see that while they find your work of interest, they have raised points that need to be addressed before we can make a decision on publication.

The referees' reports seem to be quite clear. Naturally, we will need you to address all of the points raised.

While we ask you to address all of the points raised, the following points need to be substantially worked on:

- Please add additional benchmarking studies as requested by Referee #1
- Please thoroughly discuss the limitations of your work in the discussion section of the manuscript.
- Please clarify the differences between this approach and existing similar works (both in terms of methodology and performance), and demonstrate the improvement provided by your method, ideally through experiments, where possible.

Please use the following link to submit your revised manuscript and a point-by-point response to the referees' comments (which should be in a separate document to any cover letter):

[REDACTED]

** This url links to your confidential homepage and associated information about manuscripts you may have submitted or be reviewing for us. If you wish to forward this e-mail to co-authors, please delete this link to your homepage first. **

To aid in the review process, we would appreciate it if you could also provide a copy of your manuscript files that indicates your revisions by making use of Track Changes or similar mark-up tools. Please also ensure that all correspondence is marked with your Nature Computational Science reference number in the subject line.

In addition, please make sure to upload a Word Document or LaTeX version of your text, to assist us in the editorial stage.

To improve transparency in authorship, we request that all authors identified as 'corresponding author' on published papers create and link their Open Researcher and Contributor Identifier (ORCID) with their account on the Manuscript Tracking System (MTS), prior to acceptance. ORCID helps the scientific community achieve

unambiguous attribution of all scholarly contributions. You can create and link your ORCID from the home page of the MTS by clicking on 'Modify my Springer Nature account'. For more information please visit www.springernature.com/orcid.

We hope to receive your revised paper within three weeks. If you cannot send it within this time, please let us know.

Best regards,
Kaitie

--

Kaitlin McCardle, PhD
Associate Editor
Nature Computational Science

Reviewers comments:

Reviewer #1 (Remarks to the Author):

This article presents an approach and architecture for training a graph neural network to quickly approximate DFT adsorption energy calculations of organic molecules on transition metal surfaces. To train the model the authors constructed two datasets (FG-dataset and BM-dataset) and described a specific GNN to predict the adsorption energy.

My understanding is that the authors use the representation from the relaxed structure after adsorption. This is surprising given that many of the molecules used in this work have many internal degrees of freedom and the precise adsorption configuration can change as the adsorbate relaxes on the surface. This can include rotation of bonds, and movement between different adsorption sites. Only choosing the relaxed configuration of these structures as the representation implies that there is a system to predict the relaxed adsorbate configuration, even if the graphs don't contain precise bond information. The authors should test their approach by using the unrelaxed geometries to generate the structural graphs. For systems where the adsorbate graph does not change, this should be equivalent to the current approach. Put another way, the test set should be as close as possible to the actual usage scenario, and I imagine that the authors hope to use these models to predict energies before doing DFT relaxations, not after!

My understanding is that the proposed approach disregards all information about the structure of the surfaces. This is perhaps positive to improve generalizability by focusing on the region closest to the adsorbate, but it also means that the representation can not capture any effects from strain of the underlying surface. There is a very substantial literature on how tailoring the strain effects can be a tool to engineer catalysts (including from the authors), so this should probably be

mentioned as a limitation.

The representation chosen neglects all neighbors-of-neighbors effects by truncating the graph representation at the first set of metal atoms. This is a major limitation since it means (1) the representation will be unable to differentiate between metal atoms with differing coordination, and (2) electronic effects from other elements are impossible to consider. These are both serious limitations, as there has been extensive discussions of the importance of the surface coordination in changing binding energies (e.g. Sautet et al. DOI 10.1038/nchem.2226). The model is also unable to distinguish between simple planar and stepped catalyst facets. Further, catalyst engineering by alloying elements or forming subsurface alloys are both extremely common approaches, and this approach is unable to resolve these changes. At the least these limitations should be explicitly addressed in the introduction/conclusion. The field of structure/property relationships in catalysis is quite extensive, and many other published approaches do not have these limitations.

I am particularly concerned by the lack of baseline or benchmark comparisons of this work to prior structure/property relationships, either by testing how similar approaches from the community work for this new dataset, or how the proposed model works on some of the literature datasets (e.g. anything from DOI 10.1021/acs.accounts.1c00153). This is important as the general model approach (GNNs) is extremely established in the literature (largely thanks to the work of the authors!), but the approach is presented as a new model for catalysis.

Finally, since the datasets and model both focus on the very simplest of catalytic surfaces (pure metal (111)-facets), it would be very helpful to know how much the proposed approaches improve upon scaling-relation approaches (e.g. if all data was a linear combination of simple *C, *O, *H descriptors). This would also be helpful to establish a baseline level of accuracy for the model and give some idea of the precise benefits from the more complicated machine learning model. These scaling relations tend to break down for more complex catalyst surfaces, but should work well for the simple ones in this paper. I believe all of the small molecule descriptors are known in literature for all of the surfaces in this paper.

In addition, I have a couple of minor questions:

* In section 2.4 the authors speak to the large error associated with the models predictions on aromatic compounds. I would also like to know if they have some intuition regarding the significantly lower error in sulfur-containing compounds and amides as compared to the mean.

* In section 2.5, the authors describe an MAE per non-hydrogen-atom on the "big molecules" dataset, it is unclear what the significance of this metric is, or what the reader ought to compare it to.

Reviewer #2 (Remarks to the Author):

1. What are the major claims of the paper?

The manuscript claims to have developed a graph-convolutional neural-network for the prediction of adsorption energies of organic molecules on closed-packed, ideal

metal surfaces.

2. Are they novel and do you think that they represent a significant advance in the field?

While the work is very carefully carried out and presents beautiful images, it seems to be in the direct continuation of other (not cited) works in this direction: 10.1021/acs.jpcc.7b07340 and especially 10.1038/s41467-022-29705-7

3. Is the work convincing, and if not, what further evidence would be required to strengthen the conclusions?

From my point of view, the main weakness of this (and similar) works is that the "forward" mapping (from DFT to graph) is nearly trivial, but for a fast screening, one would need to come up "de novo" with the graph. This implies that for polyfunctional molecules one would need to have an accurate estimate of which functional groups are adsorbed on the surface and which ones are not. According to my understanding, this very important aspect for practical applications is not addressed in this study. A second weak point is the restriction of investigating only closed-shell molecules, i.e., no bond-breaking, not even reactive adsorptions. These would be very relevant, especially for acids (where the dissociative adsorption mode is more stable on certain metals), but also required for being generally useful for the intended target. A third (but minor) weakness is that the hyperparameters have been optimized *after* the training of the "main" results of the paper. It would be much more convincing and coherent to present the results of the training/validation sets after the hyperparameter optimization.

4. On a more subjective note, do you feel that the paper will influence thinking in the field and be of interest to a wider computational science community of researchers? Probably not: The paper is not presenting any significantly new concept, but applies established methods to a well-known problem, which has already been addressed by similar approaches as discussed above. Nevertheless, the application is "state-of-the-art" and will certainly attract many citations.

Details:

- "DFT simulations present issues": This statement is inaccurate. The corresponding DFT computations are just costly, not an "issue".
- Section 2 should be called "Methods" rather than "Results". The actual results start in 2.4
- The choice of including Cd and Zn, but not Co and Fe is surprising. Similarly, using the Rh results as a "template" for the other metals restricts the chemical relevance compared to a more complete (and metal-dependent) adsorption mode screening, which by now can be automatized quite easily 10.1016/j.checat.2022.02.009)
- Could adsorption on alloys be described?
- I am not sure to fully understand the "Delta-ML approach": Which Delta is being learned? – According to Figure 3c, it is not the adsorption energy (which I would have expected to be learned), but some "artificial" total DFT energy. The adsorption energy is presented for the "large" molecules and it would make most sense to present this for the "small" (training/test/validation) molecules as well.
- It would be good to provide the total number of parameters that have been learned. This is important to being able to judge whether the system is over- or under-determined.
- "Finally, for the plastics, those containing single C–C and C–H bonds only best reproduced." Is not a full sentence.

- Maximum (percentage) errors should be discussed in view of Fig. 4b: They seem to be quite significant in a few cases. Therefore "excellent" is not the right adjective for qualifying the performance.
- The presented GNN does not describe "chemical processes": Just adsorption. At no point the authors have demonstrated that chemical transformations (even between closed-shell molecules) are correctly reproduced.
- The generation of the FG-dataset is said to have taken 10'000 seconds. This means less than 3 hours – for almost 3'000 DFT computations this seems wrong.

Reviewer #3 (Remarks to the Author):

This is very impressive work on the use of ML for the prediction of DFT data on adsorption energies of large molecules on transition metal surfaces. The authors have successfully implemented ML techniques and have shown that they are able to predict adsorption energies of large molecules on transition metal surfaces with the accuracy of DFT at minimal computational cost. This will be very important, in particular for the areas chosen by the authors, that is biomass and plastics. This work should be of high interest to a broad readership and I therefore recommend to accept the manuscript for publication.

Some minor (mostly technical) issues that need to be addressed:

(1) Figure 4 shows a few positive adsorption values for both DFT and GNN, but this cannot be true. Also, if dispersion is included and large molecules are considered, the adsorption should always be rather significant, but Figure 4 shows quite a few around 0.

(2) can the authors comment on the coverage effect of the dataset of adsorption energies used here. is that always at low coverages, and how would coverages be defined here (e.g. molecules per surface atom or surface area)?

(3) of all molecules considered herein, aromatics seem to be the only outlier with higher errors. This is not really surprising as the (partial) breaking of the aromaticity is very difficult to describe. This is also true for the DFT calculated data. PBE-D2 is known to have problems with the adsorption energies of aromatic compounds (e.g. benzene) on transition metal surfaces. For the purpose of ML in this contribution, the use of PBE-D2 is ok, but the authors need to comment on the shortcomings of the method when it comes to aromatics.

Author Rebuttal, second revision:

In the following the questions raised are in black, our answers in blue and the actions taken in bold.

Answer to Reviewer #1 (Remarks to the Author):

This article presents an approach and architecture for training a graph neural network to quickly approximate DFT adsorption energy calculations of organic molecules on transition metal surfaces. To train the model the authors constructed two datasets (FG-dataset and BM-dataset) and described a specific GNN to predict the adsorption energy.

We would like to thank the Reviewer for their critical analysis to our work.

My understanding is that the authors use the representation from the relaxed structure after adsorption. This is surprising given that many of the molecules used in this work have many internal degrees of freedom and the precise adsorption configuration can change as the adsorbate relaxes on the surface. This can include rotation of bonds, and movement between different adsorption sites. Only choosing the relaxed configuration of these structures as the representation implies that there is a system to predict the relaxed adsorbate configuration, even if the graphs don't contain precise bond information. The authors should test their approach by using the unrelaxed geometries to generate the structural graphs. For systems where the adsorbate graph does not change, this should be equivalent to the current approach. Put another way, the test set should be as close as possible to the actual usage scenario, and I imagine that the authors hope to use these models to predict energies before doing DFT relaxations, not after!

In the initial design of the training dataset, functional group differences were our target, thus the dataset contained the optimum structure *per* adsorbate and metal and the corresponding energy. To address the role of rotations and the different adsorption positions on the surface we have followed the heuristics developed in our group (DOIs: 10.1021/jp502819s, 10.1021/cs501698w) and **summarized in Methods Density Functional Theory and SI Note S2 Adsorption Conformational Search**. Particularly, methyl groups are known to present 2 H atoms close to the surface instead of one directly pointing towards it as it reduces the repulsion to the surface. In any case the training set consists of small molecules many of them with rigid bonds, and from the gas-phase point of view the maximum deviation is 0.4 eV. As for the surface adsorption sites to demonstrate that different adsorption modes (fcc, hcp, top, bridge) lead to different graphs and GNN predictions, we randomly selected two molecules for each chemical family (18 molecules in total) and placed the center of mass of the molecules at a fixed distance into two different surface positions on a random metal, generated the graph from the unrelaxed geometry, estimated the GNN prediction and compared it to the relaxed DFT energy. The results of this test show that in all cases the generated graph is different depending on the adsorption site, generally following the same relative energy ordering. **This point has been added to the Supporting Information, see S11-S13.**

Regarding the input for GAME-Net, we are now clarifying in the main text that to obtain the graph and training the model, we have employed the optimized structures from the explicit DFT calculations in the FG-dataset (3,315 samples only (111) facet). Once the GNN is trained, an algorithm that docks the molecule at a reasonable distance on any of the investigated metal surfaces

can be employed to generate the first graph (unrelaxed) and an estimate of the energy can be obtained from GAME-Net. **This is now properly described in the main manuscript, Discussion section.**

My understanding is that the proposed approach disregards all information about the structure of the surfaces. This is perhaps positive to improve generalizability by focusing on the region closest to the adsorbate, but it also means that the representation cannot capture any effects from strain of the underlying surface. There is a very substantial literature on how tailoring the strain effects can be a tool to engineer catalysts (including from the authors), so this should probably be mentioned as a limitation.

We fully agree with the Reviewer that structure sensitivity, i.e. the dependence of the adsorption energy to different facet orientations is an important aspect. We have performed a full DFT evaluation of the 207 molecules in the FG-dataset on the (100) and (110) facets of the fcc-metals and compared to the results from GAME-Net (trained just with (111) and the other lowest surfaces (110) bcc and (0001) hcp). The results are remarkable, as the mean absolute error for the system energy is 0.34 eV for the (100) subset and 0.41 eV for the (110) samples. **The data is now added to the ioChem-BD database, and a comment on this aspect has been added to the Discussion.** As for the comment on the selection of the surface model “by focusing on the region closest to the adsorbate,” extensive benchmark demonstrates the relevance of the ensemble approach, see point on benchmarks (4) in this answer. Other more complex aspects such as strain could induce changes in the adsorption energies that cannot be captured by the simplicity of our graph representation. As for the dopant effects, the electronic ones would likely be incorporated while those implying strain would be less effectively described. **We are now indicating this in our list of potential limitations of the method in the Discussion Section.**

The representation chosen neglects all neighbors-of-neighbors effects by truncating the graph representation at the first set of metal atoms. This is a major limitation since it means (1) the representation will be unable to differentiate between metal atoms with differing coordination, and (2) electronic effects from other elements are impossible to consider. These are both serious limitations, as there has been extensive discussions of the importance of the surface coordination in changing binding energies (e.g. Sautet et al. DOI 10.1038/nchem.2226). The model is also unable to distinguish between simple planar and stepped catalyst facets. Further, catalyst engineering by alloying elements or forming subsurface alloys are both extremely common approaches, and this approach is unable to resolve these changes. At the least these limitations should be explicitly addressed in the introduction/conclusion. The field of structure/property relationships in catalysis is quite extensive, and many other published approaches do not have these limitations.

We agree with the Reviewer that this might be seen as a limitation of the method. However, GAME-Net fitting on the FG-dataset for the (111) and equivalent facets can reproduce the DFT adsorption energies of these open surfaces. This happens because for open or stepped surfaces the graphs are different as **we are now demonstrating in Figures S14-15. This is explicitly shown**

in the manuscript section Model Performance and the added the reference suggested. As for the other sources of distortion of the surface we are now indicating a list of potential limitations to the method in the second to last paragraph.

I am particularly concerned by the lack of baseline or benchmark comparisons of this work to prior structure/property relationships, either by testing how similar approaches from the community work for this new dataset, or how the proposed model works on some of the literature datasets (e.g. anything from DOI 10.1021/acs.accounts.1c00153). This is important as the general model approach (GNNs) is extremely established in the literature (largely thanks to the work of the authors!), but the approach is presented as a new model for catalysis.

We agree with the Reviewer that standardization and shared datasets and validation via application of competing techniques. We have carried out an extensive benchmark on data and algorithms.

GAME-Net with data from the literature: The proposed datasets mainly comprise CH_x and O-related fragments. We employed data from Andersen et al. (DOI 10.1021/acscatal.8b04478) and two from our previous research (DOI 10.1039/d1cy01423d and 10.1038/s41467-019-12709-1). The dataset from Andersen is generated with a different DFT code (Quantum Espresso vs. VASP) and functional, and contains only 6 fragments with maximum two atoms (up to C₁ relevant to methanation and oxygen evolution reaction). This is the worst possible situation for GAME-Net, as the FG-dataset employed by us does not contain any fragment. The results show that the performance is relatively poor (MAE=1.6 eV). Instead, when the same DFT settings are used, like in our PCA-database (DOI 10.1038/s41467-019-12709-1) containing fragments with C and O (up to two carbon atoms), GAME-Net yields a MAE=0.85 eV, and if larger fragments, like those found in the CO₂ to propylene study are considered, we obtain a lower MAE, of 0.58 eV. This error is one third of the value obtained with the Andersen dataset. The benchmarks are thus very consistent, demonstrating that refitting is needed when changing functional, and explicit inclusion of fragments would be required to improve GAME-Net results for datasets of reaction intermediates. **The results for this benchmark have been added in the Discussion section of the Manuscript and in the Supplementary Information, Note S9, Figures S19-S22, and Table S14.**

GAME-Net vs other models: We have employed our FG-dataset to compare GAME-Net with the models that participated in the Open Catalyst Project (OCP). We have used two out of the four: DimeNet++ and PaiNN with the OCP optimized hyperparameters. The original target of these two models is to start from an initial molecular structure and the full metal slab and retrieve the optimized corresponding adsorption energy. We devised a strategy aiming at investigating the role of the initial vs. final geometry, the number of metal atoms included in the graph representation (ensemble vs. full slab). A total of 8 benchmark models were created for each of them. We performed the same cross validation recipe presented in the main manuscript for GAME-Net and predicted the BM dataset. **The results for this benchmark have been added in the Discussion section of the Manuscript and in the Supplementary Information, Note S10, Figures S23-24, and Table S15-S16.**

The main take home messages from the benchmark are:

- Our FG-dataset is robust, diverse, chemically meaningful, and well-balanced. This allows all models to provide reasonable results on the large chemical space of the BM-dataset.
- GAME-Net performs similarly to DimeNet++ and PaiNN at a much lower computational cost and time (needing a GPU laptop vs. a GPU cluster, about 1-2 orders of magnitude).
- GAME-Net graph representation based on the surface ensemble is always better than the full slab representation (typically used by the other GNNs). Notice, that in these models, the metal centers might be represented by many more attributes (atom type, positions 3D structure, forces) than in our case.
- While most of the GNNs employ an initial structure geometry (3D) to provide a final optimized energy, GAME-Net is more intuitive and interpretable as the guess structures are distilled as graphs and thus the chemistry become completely transparent in the interpretation of the bonds at the interface. This is particularly evident when assessing the adsorption energy on different surface positions (top vs. hollow sites) where we can trace completely the type of bond and this information is fundamental to experimental chemist to understand reactivity. For instance, hydrogenation is known to lead to different products for alkenes when adsorbed in on top positions compared to bridge reactivity. In GAME-Net this result can be easily obtained.

A summary of these benchmarks has been added to the main text in the Discussion section and all the data is presented in the SI.

Finally, since the datasets and model both focus on the very simplest of catalytic surfaces (pure metal (111)-facets), it would be very helpful to know how much the proposed approaches improve upon scaling-relation approaches (e.g. if all data was a linear combination of simple *C, *O, *H descriptors). This would also be helpful to establish a baseline level of accuracy for the model and give some idea of the precise benefits from the more complicated machine learning model. These scaling relations tend to break down for more complex catalyst surfaces, but should work well for the simple ones in this paper. I believe all of the small molecule descriptors are known in literature for all of the surfaces in this paper.

Linear-scaling relationships (LSR) were devised for partially hydrogenated fragments, i.e. OH will scale well with O but H₂O would be almost constant for different metals. In our case all considered molecules are closed-shells, closer to the water case. Still, we have compared our model to the LSR of Pedersen et al. (DOI 10.1103/PhysRevLett.99.016105). Unfortunately, all the adsorbates considered in this pioneering work have only one functional group with the exception of cysteine, for which the adsorption reported is for the zwitterionic configuration (i.e. only interacting with the thiolate). When we place cysteine on the fcc (111) metals, we observe that the graph metal contacts to more than one functionality simultaneously at least for the most reactive metals and might be too simple for large molecules. **We are including this discussion in Figure S25.**

Figure S25. Linear-scaling can provide good models complex molecules like cysteine (zwitterionic form) if they interact only with one atom, S. However, if the molecule interacts with several functional groups the model might be too simplistic. The approach is very good for the coinage metals, but for reactive metals more than one interaction exist, and the adsorption energy deviates.

In addition, I have a couple of minor questions:

* In section 2.4 the authors speak to the large error associated with the models predictions on aromatic compounds. I would also like to know if they have some intuition regarding the significantly lower error in sulfur-containing compounds and amides as compared to the mean.

We analyzed the distribution of the DFT energy, the sulfur and amide sets span a similar energy window with similar distribution (4 peaks). This could be one of reasons for the GAME-Net differential performance. Additionally, the cyclic nature of the aromatic molecules in our dataset is not specifically encoded in GAME-Net, and significantly none of the other compared nets including the 3D structures DimeNet++ and PaiNN improve this situation, **this aspect has been added to the Model Performance.**

Figure: Energy span for the mentioned functional groups, showing the largest energy window for aromatics.

* In section 2.5, the authors describe an MAE per non-hydrogen-atom on the “big molecules” dataset, it is unclear what the significance of this metric is, or what the reader ought to compare it to.

We have changed the MAE and now it is obtained in a per-atom (hydrogen included) way in the molecule, giving a value of 0.016 eV/atom.

Answer to Reviewer #2 (Remarks to the Author):

1. What are the major claims of the paper? The manuscript claims to have developed a graph-convolutional neural-network for the prediction of adsorption energies of organic molecules on closed-packed, ideal metal surfaces.

Exactly, this is the major point of the manuscript and has been clarified in the Abstract, Introduction and Discussion sections.

2. Are they novel and do you think that they represent a significant advance in the field? While the work is very carefully carried out and presents beautiful images, it seems to be in the direct continuation of other (not cited) works in this direction: 10.1021/acs.jpcc.7b07340 and especially 10.1038/s41467-022-29705-7

We apologize for not including these works that indeed are very meaningful. In the first suggestion, a group additivity model was presented for 200 species, basically ethanol to C₄ alcohols, both as gas-phase and adsorbed fragments with remarkable 0.12 eV errors. In the second, the species investigated contain up to 6 C or O atoms and graph-theory, force fields, explicit DFT calculations and ML are employed to analyze the stability of different configurations on 11 metals. Again, remarkable results in terms of efficiency are obtained.

In our case, we have broadened the chemical space by orders of magnitude by creating a robust, chemically meaningful dataset, including all the relevant functional groups in chemistry. Moreover, we have relaxed the additivity constraint that might be relevant for molecules with interacting functional groups. The references have been added in the Introduction and Discussion section, and a critical comparison showing the need for complete use of the chemical space in advanced applications has been added.

3. Is the work convincing, and if not, what further evidence would be required to strengthen the conclusions?

From my point of view, the main weakness of this (and similar) works is that the “forward” mapping (from DFT to graph) is nearly trivial, but for a fast screening, one would need to come up “de novo” with the graph. This implies that for polyfunctional molecules one would need to

have an accurate estimate of which functional groups are adsorbed on the surface and which ones are not. According to my understanding, this very important aspect for practical applications is not addressed in this study.

We agree with the Reviewer that starting from an already optimized (CONTCAR-like) structure obtaining the graph and then the energy might be seen as trivial. However, if we use a standardized software to place the molecule on the surface (e.g., ASE, DockOnSurf) and we generate the initial structure (POSCAR-like), GAME-Net has enough information to indicate if an interaction between two nodes in the graph (for instance a H atom from a terminal methyl group) and the surface (Au for instance) is very weak. Then the corresponding adsorption energy contribution from this subgraph will be zero. Our method is so fast that we can afford to screen all the possibilities given by the docking approaches and take the lowest energy ones. This is actually an advantage of our method. **This analysis has been added to the Model Performance section of the manuscript.**

A second weak point is the restriction of investigating only closed-shell molecules, i.e., no bond-breaking, not even reactive adsorptions. These would be very relevant, especially for acids (where the dissociative adsorption mode is more stable on certain metals), but also required for being generally useful for the intended target.

As mentioned in the answer to Reviewer 1 (comment 4), we benchmark GAME-Net with datasets containing only open-shell fragments. Notice that all previous studies in the field are addressed to dealing with fragments with less than 4 carbon atoms. **The results of the benchmarks are summarized in Note S9 and demonstrate that the model works reasonably considering the training set employed.** We agree that the long-time goal of the project should be to present the dissociative energies and eventually reaction network. Having said that, a benchmark on the non-reacted species is the first mandatory approach to find a robust and transferable solution. **Our approach shows that it is indeed possible to evaluate very large molecules at a very low cost.**

A third (but minor) weakness is that the hyperparameters have been optimized *after* the training of the “main” results of the paper. It would be much more convincing and coherent to present the results of the training/validation sets after the hyperparameter optimization.

We agree with the Reviewer. Since we have added more data to the FG-dataset (molecules on Fe (110) and Co (0001)), we have followed the Reviewer’s advice and **all the results presented in the manuscript come from the final optimized model architecture, having performed the hyperparameter optimization as first step, subsection Model Architecture.**

4. On a more subjective note, do you feel that the paper will influence thinking in the field and be of interest to a wider computational science community of researchers?

Probably not: The paper is not presenting any significantly new concept, but applies established methods to a well-known problem, which has already been addressed by similar approaches as discussed above. Nevertheless, the application is “state-of-the-art” and will certainly attract many citations.

The project demonstrates that it is possible to **generate a robust, chemically diverse and meaningful dataset from which the model learns the chemistry** to build sound, powerful and cheap models for the estimation of binding energies and that can be employed by any experimental group agnostic to the use of DFT. Thus, we show that there is a clear way to go from small molecules with different functional groups to large species containing more than one of these functionalities even at short interacting distances. This was not addressed previously. **In addition, the GAME-Net can be employed to get a fast estimate of the adsorption of different competing structures.** Something that can be very valuable at a very low computational cost.

Details:

- “DFT simulations present issues”: This statement is inaccurate. The corresponding DFT computations are just costly, not an “issue”.

Corrected.

- Section 2 should be called “Methods” rather than “Results”. The actual results start in 2.4
The naming of the section follows the Author Guidelines provided by Nature.

- The choice of including Cd and Zn, but not Co and Fe is surprising. Similarly, using the Rh results as a "template" for the other metals restricts the chemical relevance compared to a more complete (and metal-dependent) adsorption mode screening, which by now can be automatised quite easily (10.1016/j.checat.2022.02.009)

We have added Co and Fe to the FG-dataset, demonstrating that the same GAME-Net architecture work in these cases. We have added the reference suggested in the Supplementary Information Note S3.

- Could adsorption on alloys be described?

This discussion has been added following also the suggestion of Reviewer 1 to the Discussion section where the potential and limitations of the method are described. While we think electronic terms could be easily introduced the role of geometric distortions like local strain would be more difficult to address with such a simplified model.

- I am not sure to fully understand the “Delta-ML approach”: Which Delta is being learned? – According to Figure 3c, it is not the adsorption energy (which I would have expected to be learned), but some “artificial” total DFT energy. The adsorption energy is presented for the “large” molecules and it would make most sense to present this for the “small” (training/test/validation) molecules as well.

The Delta learnt is the change in the energy of the molecule due to the interaction with the metal. This is not a physically measurable parameter but from the GNN optimization standpoint turns out to simplify the complexity even for the other architectures compared in the benchmark. To get the adsorption energy, one should subtract from this energy the gas-phase energy of the molecule. As the nested-cross validation performed to assess the model performance involves training the

GAME-Net multiple times (leading to 20 models with same architecture but different parameters), we compared the adsorption energy prediction using the final model trained with the whole FG-dataset (no test set), the same model used for the BM-dataset. For the comparison, the gas-phase energy is taken from both GAME-Net and DFT. **The new results have been added to the Supplementary Information, Figure S8-S9.**

- It would be good to provide the total number of parameters that have been learned. This is important to being able to judge whether the system is over- or under-determined.

GAME-Net has 285,761 parameters. **We have added this detail to the Model Architecture subsection and in Note S6.**

- “Finally, for the plastics, those containing single C–C and C–H bonds only best reproduced.” Is not a full sentence.

We have fixed the sentence.

- Maximum (percentage) errors should be discussed in view of Fig. 4b: They seem to be quite significant in a few cases. Therefore “excellent” is not the right adjective for qualifying the performance.

We are indicating that in most of the cases it is very good. **We modified our claims to be more specific in the last paragraph of the Section Application to Industrially relevant problems.**

- The presented GNN does not describe “chemical processes”: Just adsorption. At no point have the authors demonstrated that chemical transformations (even between closed-shell molecules) are correctly reproduced.

We have fixed this issue.

- The generation of the FG-dataset is said to have taken 10'000 seconds. This means less than 3 hours – for almost 3'000 DFT computations this seems wrong.

We have fixed this issue in the Discussion section. The total elapsed time needed to generate the FG-dataset (3,315 samples) used to train GAME-Net has been 16,617 hours, leading to an average of 5.5 hours per run.

Answer Reviewer #3 (Remarks to the Author):

This is very impressive work on the use of ML for the prediction of DFT data on adsorption energies of large molecules on transition metal surfaces. The authors have successfully implemented ML techniques and have shown that they are able to predict adsorption energies of

large molecules on transition metal surfaces with the accuracy of DFT at minimal computational cost. This will be very important, particularly for the areas chosen by the authors, that is biomass and plastics. This work should be of high interest to a broad readership, and I therefore recommend to accept the manuscript for publication.

We would like to thank the Reviewer for their comments.

Some minor (mostly technical) issues that need to be addressed:

(1) Figure 4 shows a few positive adsorption values for both DFT and GNN, but this cannot be true. Also, if dispersion is included and large molecules are considered, the adsorption should always be rather significant, but Figure 4 shows quite a few around 0.

Endothermic values can appear because of the structural changes to make the bond at the surface, see the answer to Reviewer #1 comment 1. The message is that adsorption in these structures is weak. **We are now describing better this point in the Discussion.**

(2) can the authors comment on the coverage effect of the dataset of adsorption energies used here. is that always at low coverages, and how would coverages be defined here (e.g. molecules per surface atom or surface area)?

This is an important point, and **it is now added to the subsection Density Functional Theory.** The studied adsorption corresponds to the isolated adsorbed molecules. The samples in the FG-dataset present a surface coverage around 0.02 molecules/ \AA^2 (1 molecule per 50 \AA^2).

(3) of all molecules considered herein, aromatics seem to be the only outlier with higher errors. This is not really surprising as the (partial) breaking of the aromaticity is very difficult to describe. This is also true for the DFT calculated data. PBE-D2 is known to have problems with the adsorption energies of aromatic compounds (e.g. benzene) on transition metal surfaces. For the purpose of ML in this contribution, the use of PBE-D2 is ok, but the authors need to comment on the shortcomings of the method when it comes to aromatics.

The D2 values correspond to our own refitted values for metals, see DOI 10.1021/ct5006467, where we had the following Table which explicitly considers the benzene case. **We have updated the reference to our work and remarked this point in the Computational Details.**

Table 2. Binding Energies (BE, eV) for the Adsorption of Benzene on Different Metal Surfaces^a

metal	PBE	DFT-D2	DFT-D3	DFT-vdW ^{surf} (37)	this work	exp
M(4 × 4) + 2C ₆ H ₆						
Pd	0.84	2.58	1.91		1.88	1.64 (31)–1.83 (50)
Pt	0.59	3.04	1.99		1.57	1.54 (52)–1.60 (51) (1.54) (57)
M(3 × 3) + C ₆ H ₆						
Pd	1.22	2.82	2.22	2.14	2.17	
Pt	0.97	3.23	2.27	1.96	1.92	(1.94) (57)
Cu	0.14	0.91	1.02	0.86	0.47	0.70 (53)–0.71 (38)
Ag	0.07	0.90	0.77	0.75	0.41	0.46 (58)–0.45 (54)
Au	0.05	1.35	0.88	0.74	0.48	0.63 (55)

^aThe experimental values estimated from TPD data are reported via an entropy-corrected prefactor. (56) For C₆H₆/Pt(111) we report microcalorimetric measurements in parentheses. The Ag data (54, 58) corresponds to the shoulder that appears at the same coverage as our calculations were performed, 140 K. There is another higher temperature peak at lower coverages 205 K. The TPD data from Au (55) shows a high energy peak that continuously shifts from 239 to 175 K, thus the assignment is unclear.

Decision Letter, second revision:

Date: 6th March 23 16:24:10
Last Sent: 6th March 23 16:24:10
Triggered By: Kaitlin McCardle
From: kaitlin.mccardle@us.nature.com
To: nlopez@iciq.es
CC: computationalscience@nature.com
Subject: AIP Decision on Manuscript NATCOMPUTSCI-22-1074B
Message: Our ref: NATCOMPUTSCI-22-1074B

6th March 2023

Dear Dr. López,

Thank you for submitting your revised manuscript "Fast Evaluation of the Adsorption Energy of Organic Molecules on Metals via Graph Neural Networks" (NATCOMPUTSCI-22-1074B). It has now been seen by the original referees and their comments are below. The reviewers find that the paper has improved in revision, and therefore we'll be happy in principle to publish it in Nature Computational Science, pending minor revisions to satisfy the referees' final requests and to comply with our editorial and formatting guidelines.

We are now performing detailed checks on your paper and will send you a checklist detailing our editorial and formatting requirements in about a week. Please do not

upload the final materials and make any revisions until you receive this additional information from us.

TRANSPARENT PEER REVIEW

Nature Computational Science offers a transparent peer review option for original research manuscripts. We encourage increased transparency in peer review by publishing the reviewer comments, author rebuttal letters and editorial decision letters if the authors agree. Such peer review material is made available as a supplementary peer review file. **Please state in the cover letter 'I wish to participate in transparent peer review' if you want to opt in, or 'I do not wish to participate in transparent peer review' if you don't.** Failure to state your preference will result in delays in accepting your manuscript for publication. Please note: we allow redactions to authors' rebuttal and reviewer comments in the interest of confidentiality. If you are concerned about the release of confidential data, please let us know specifically what information you would like to have removed. Please note that we cannot incorporate redactions for any other reasons. Reviewer names will be published in the peer review files if the reviewer signed the comments to authors, or if reviewers explicitly agree to release their name. For more information, please refer to our [FAQ page](https://www.nature.com/documents/nr-transparent-peer-review.pdf).

Thank you again for your interest in Nature Computational Science Please do not hesitate to contact me if you have any questions.

Sincerely,

Kaitlin McCardle
Editor
Nature Computational Science

ORCID

Reviewer #1 (Remarks to the Author):

The authors have addressed my comments with the additional benchmarks and comparisons. I am also happy to see the detailed discussion about the limitations and pros/cons of various literature approaches for this new model!

Reviewer #2 (Remarks to the Author):

The authors have well replied to my initial concerns and I am overall satisfied with the revision from this point of view. My only remaining small comment is the following:

The authors use 3315 energy computations to fit 285'761 parameters. I would like to at least see a comment on this in the main text: there must be redundant/useless/ill-defined parameters as only the energy (not the forces) are learned. It is probably still the most practical approach to move on, but it should be discussed and acknowledged.

The main limitation of their work that they are not able to simply address is the restriction to closed-shell molecules. Nevertheless, one can indeed argue that their work is a necessary first step.

I am much less convinced by the replies of the authors to the (valid) concerns and remarks of reviewer #1. However, I presume that he/she will judge for him/herself.

Reviewer #3 (Remarks to the Author):

The authors addressed my concerns satisfactorily. In my opinion, this study presents a very hot topic and should pave the way for future studies on the functionalization of large molecules (e.g. biomass, plastics) on transition metal surfaces and therefore deserves publication.

Author rebuttal, third revision:

Answer to Reviewer #1:

The authors have addressed my comments with the additional benchmarks and comparisons. I am also happy to see the detailed discussion about the limitations and pros/cons of various literature approaches for this new model!

We are grateful for the new comments of the Reviewer and they positive evaluation of our revision.

Answer to Reviewer #2:

The authors have well replied to my initial concerns and I am overall satisfied with the revision from this point of view.

We would like to thank the Reviewer again for they detailed evaluation of our work that has helped us improving the quality of the initial manuscript.

My only remaining small comment is the following:

The authors use 3315 energy computations to fit 285'761 parameters. I would like to at least see a comment on this in the main text: there must be redundant/useless/ill-defined parameters as only

the energy (not the forces) are learned. It is probably still the most practical approach to move on, but it should be discussed and acknowledged.

We have added a comment in the Methods Section “GAME-Net architecture” regarding the number of data needed to fit the model. Now the manuscript reads “**GAME-Net has been trained with 2853 graphs, ending up with a GNN that contains 285,761 trainable parameters, 129,121 of them (45%) belonging to the GMT pooling layer due to its internal complexity, and the remaining parameters equally distributed among the other layers. Likely, the number of parameters employed in the GNN contains some redundancies, however, eliminating those might be a more complex task than employing the very compact structure of GAME-Net.**”

The main limitation of their work that they are not able to simply address is the restriction to closed-shell molecules. Nevertheless, one can indeed argue that their work is a necessary first step.

We are aiming at continuing with the project in the near future adding open-shell fragments.

I am much less convinced by the replies of the authors to the (valid) concerns and remarks of reviewer #1. However, I presume that he/she will judge for him/herself.

We are grateful that Reviewer #1 positively evaluated the new version of our manuscript.

Answer to Reviewer #3

The authors addressed my concerns satisfactorily. In my opinion, this study presents a very hot topic and should pave the way for future studies on the functionalization of large molecules (e.g. biomass, plastics) on transition metal surfaces and therefore deserves publication.

We are very grateful of the positive view expressed by the Reviewer.

Final Decision Letter:

Date: 23rd March 23 11:42:35

Last Sent: 23rd March 23 11:42:35

Triggered By: Kaitlin McCardle

From: kaitlin.mccardle@us.nature.com

To: nlopez@iciq.es

BCC: rjsart@springernature.com,computationalscience@nature.com,fernando.chirigati@us.nature.com,kaitlin.mccardle@us.nature.com,rjsproduction@springernature.com

Subject : Decision on Nature Computational Science manuscript NATCOMPUTSCI-22-1074C

Message: Dear Professor López,

We are pleased to inform you that your Article "Fast Evaluation of the Adsorption Energy of Organic Molecules on Metals via Graph Neural Networks" has now been accepted for publication in Nature Computational Science.

Once your manuscript is typeset, you will receive an email with a link to choose the appropriate publishing options for your paper and our Author Services team will be in touch regarding any additional information that may be required.

Please note that *Nature Computational Science* is a Transformative Journal (TJ). Authors may publish their research with us through the traditional subscription access route or make their paper immediately open access through payment of an article-processing charge (APC). Authors will not be required to make a final decision about access to their article until it has been accepted. [Find out more about Transformative Journals](https://www.springernature.com/gp/open-research/transformative-journals)

Acceptance of your manuscript is conditional on all authors' agreement with our publication policies (see <https://www.nature.com/natcomputsci/for-authors>). In particular your manuscript must not be published elsewhere and there must be no announcement of the work to any media outlet until the publication date (the day on which it is uploaded onto our web site).

Before your manuscript is typeset, we will edit the text to ensure it is intelligible to our wide readership and conforms to house style. We look particularly carefully at the titles of all papers to ensure that they are relatively brief and understandable.

Once your manuscript is typeset and you have completed the appropriate grant of rights, you will receive a link to your electronic proof via email with a request to make any corrections within 48 hours. If, when you receive your proof, you cannot meet this deadline, please inform us at rjsproduction@springernature.com immediately.

If you have queries at any point during the production process then please contact the production team at rjsproduction@springernature.com. Once your paper has been scheduled for online publication, the Nature press office will be in touch to confirm the details.

Content is published online weekly on Mondays and Thursdays, and the embargo is set at 16:00 London time (GMT)/11:00 am US Eastern time (EST) on the day of publication. If you need to know the exact publication date or when the news embargo will be lifted, please contact our press office after you have submitted your proof corrections. Now is the time to inform your Public Relations or Press Office about your paper, as they might be interested in promoting its publication. This will allow them time to prepare an accurate and satisfactory press release. Include your manuscript tracking number NATCOMPUTSCI-22-1074C and the name of the journal, which they will need when they contact our office.

About one week before your paper is published online, we shall be distributing a press release to news organizations worldwide, which may include details of your work. We are happy for your institution or funding agency to prepare its own press release, but it must mention the embargo date and Nature Computational Science. Our Press Office will contact you closer to the time of publication, but if you or your Press Office have any inquiries in the meantime, please contact press@nature.com.

We welcome the submission of potential cover material (including a short caption of around 40 words) related to your manuscript; suggestions should be sent to Nature Computational Science as electronic files (the image should be 300 dpi at 210 x 297 mm in either TIFF or JPEG format). We also welcome suggestions for the Hero Image, which appears at the top of our [home page](http://www.nature.com/natcomputsci); these should be 72 dpi at 1400 x 400 pixels in JPEG format. Please note that such pictures should be selected more for their aesthetic appeal than for their scientific content, and that colour images work better than black and white or grayscale images. Please do not try to design a cover with the Nature Computational Science logo etc., and please do not submit composites of images related to your work. I am sure you will understand that we cannot make any promise as to whether any of your suggestions might be selected for the cover of the journal.

Best regards,

Kaitlin McCardle, PhD
Associate Editor
Nature Computational Science

P.S. Click on the following link if you would like to recommend Nature Computational Science to your librarian: https://www.springernature.com/gp/librarians/recommend-to-your-library

** Visit the Springer Nature Editorial and Publishing website at www.springernature.com/editorial-and-publishing-jobs for more information about our career opportunities. If you have any questions please click here.**

RightsLink

Home

Help ▾

Live Chat

Sign in

Create Account

Costless Derivation of Dispersion Coefficients for Metal Surfaces

Author: Neyvis Almora-Barrios, Giuliano Carchini, Piotr Błoński, et al

Publication: Journal of Chemical Theory and Computation

Publisher: American Chemical Society

Date: Nov 1, 2014

Copyright © 2014, American Chemical Society

PERMISSION/LICENSE IS GRANTED FOR YOUR ORDER AT NO CHARGE

This type of permission/license, instead of the standard Terms and Conditions, is sent to you because no fee is being charged for your order. Please note the following:

- Permission is granted for your request in both print and electronic formats, and translations.
- If figures and/or tables were requested, they may be adapted or used in part.
- Please print this page for your records and send a copy of it to your publisher/graduate school.
- Appropriate credit for the requested material should be given as follows: "Reprinted (adapted) with permission from {COMPLETE REFERENCE CITATION}. Copyright {YEAR} American Chemical Society." Insert appropriate information in place of the capitalized words.
- One-time permission is granted only for the use specified in your RightsLink request. No additional uses are granted (such as derivative works or other editions). For any uses, please submit a new request.

If credit is given to another source for the material you requested from RightsLink, permission must be obtained from that source.

BACKCLOSE WINDOW